# Epigenetic Priming with Decitabine Augments the Therapeutic Effect of Cisplatin on Triple-Negative Breast Cancer Cells through Induction of Proapoptotic Factor NOXA

**DOI:** 10.3390/cancers14010248

**Published:** 2022-01-04

**Authors:** Wataru Nakajima, Kai Miyazaki, Masahiro Sakaguchi, Yumi Asano, Mariko Ishibashi, Tomoko Kurita, Hiroki Yamaguchi, Hiroyuki Takei, Nobuyuki Tanaka

**Affiliations:** 1Department of Molecular Oncology, Institute for Advanced Medical Sciences, Nippon Medical School, Tokyo 113-0033, Japan; nakaji@nms.ac.jp (W.N.); s16-098mk@nms.ac.jp (K.M.); m-sakaguchi@nms.ac.jp (M.S.); yumi-nit@nms.ac.jp (Y.A.); 2Department of Hematology, Nippon Medical School, Tokyo 113-0033, Japan; hiroki@nms.ac.jp; 3Department of Microbiology and Immunology, Nippon Medical School, Tokyo 113-0033, Japan; mariko-ishibashi@nms.ac.jp; 4Department of Breast Surgery and Oncology, Nippon Medical School, Tokyo 113-0033, Japan; tomoko28@nms.ac.jp (T.K.); takei-hiroyuki@nms.ac.jp (H.T.)

**Keywords:** decitabine, cisplatin, triple-negative breast cancer, DNA methylation, drug combination, apoptosis, NOXA

## Abstract

**Simple Summary:**

Triple-negative breast cancer is a subset of breast cancer that occurs frequently in young women and tends to exhibit aggressive, metastatic behavior. The therapeutic molecular targets found in other types of breast cancer are absent; therefore, this type of cancer has a poorer prognosis. To search for effective treatments for this type of cancer, we analyzed the effect of the DNA-demethylating agent, decitabine, which is commonly used in patients with myelodysplastic syndrome. We found that in triple-negative breast cancer cell subtypes, inhibition of cell death and cell growth in response to cisplatin, which is used to treat metastatic breast cancer, is enhanced when used in combination with decitabine. We also found that in decitabine-refractory cell subtypes, cisplatin alone is effective at inducing cell death. These results indicate the possibility of effective new combination therapies in triple-negative breast cancers.

**Abstract:**

Epigenetic alterations caused by aberrant DNA methylation have a crucial role in cancer development, and the DNA-demethylating agent decitabine, is used to treat hematopoietic malignancy. Triple-negative breast cancers (TNBCs) have shown sensitivity to decitabine; however, the underlying mechanism of its anticancer effect and its effectiveness in treating TNBCs are not fully understood. We analyzed the effects of decitabine on nine TNBC cell lines and examined genes associated with its cytotoxic effects. According to the effect of decitabine, we classified the cell lines into cell death (D)-type, growth inhibition (G)-type, and resistant (R)-type. In D-type cells, decitabine induced the expression of apoptotic regulators and, among them, NOXA was functionally involved in decitabine-induced apoptosis. In G-type cells, induction of the cyclin-dependent kinase inhibitor, p21, and cell cycle arrest were observed. Furthermore, decitabine enhanced the cytotoxic effect of cisplatin mediated by NOXA in D-type and G-type cells. In contrast, the sensitivity to cisplatin was high in R-type cells, and no enhancing effect by decitabine was observed. These results indicate that decitabine enhances the proapoptotic effect of cisplatin on TNBC cell lines that are less sensitive to cisplatin, indicating the potential for combination therapy in TNBC.

## 1. Introduction

Triple-negative breast cancers (TNBCs) are defined as tumors that lack expression of estrogen receptor (ER), progesterone receptor (PR), and human epidermal growth factor receptor type 2 (HER2) [1]. TNBCs represent approximately 16% of breast cancer cases [2], occur frequently in young women, and tend to exhibit aggressive, metastatic behavior [3]. Because therapeutic molecular targets found in other type of breast cancers, such as ER, PR, or HER2, are not expressed, TNBCs show a poorer prognosis caused by their high propensity for metastatic progression [1,3]. Therefore, conventional chemotherapy, individually or in combination with surgery and/or radiotherapy, is the current standard treatment for TNBC [4]. TNBCs show some sensitivity to chemotherapy, particularly to cytotoxic agents, such as microtubule stabilizers (taxanes), anthracyclines (doxorubicin and epirubicin), and platinum agents (carboplatin and cisplatin). At present, taxane- and anthracycline-based chemotherapy is the mainstay of treatment of early TNBCs, and several clinical trials involving the addition of platinum agents have shown improvement of pathological response rates [5]. Moreover, treatment with these drugs is part of the standard therapy for high-risk patients, for example, patients with lymph node metastasis [5,6].

Sensitivity of TNBCs to DNA-damaging compounds, such as platinum agents, is thought to result from the presence of DNA repair defects [6], for example, mutations in the DNA-damage response regulators *BRCA1/2* (present in approximately 25% of TNBCs [7]) and *TP53* (in approximately 80% of TNBCs [8]). However, intrinsic or acquired resistance against platinum treatment limits their clinical application for TNBC [9,10]. Cisplatin is a commonly used platinum-based compound that cross-links purine bases in DNA, causing DNA-damage responses [11]. Numerous studies have revealed that the cytotoxic effect of cisplatin is mainly caused by DNA-damage induced apoptosis [12]. Evading apoptosis is one of the hallmarks of cancer [13,14] and, in addition to altered pharmacokinetic factors such as drug metabolism and elimination, impaired induction of apoptosis is an important cause of resistance against cisplatin and other platinum-based drugs [10,15]. Indeed, deregulation of apoptosis-regulating BCL2 family proteins has been observed in many types of cancer, including breast cancers. These changes include overexpression of genes encoding antiapoptotic BCL-XL and/or MCL1 and inactivating mutations or epigenetic inactivation of genes encoding proapoptotic BAX, BAK, or BAX/BAK activators that are BCL-2 homology 3 (BH3)-only members of the BCL2 family proteins, such as BIM, PUMA, or NOXA [15,16,17].

Epigenetic alterations involving DNA methylation and histone modifications have a crucial role in the oncogenesis, tumor development, and chemotherapeutic resistance of human cancers [18,19] and are therefore targets of cancer chemotherapy [20]. In eukaryotic genomes, methylation of cytosines within CpG dinucleotides is involved in transcriptional silencing and maintaining and regulating chromatin structure, genome stability, and cell fate [21,22]. Four DNA cytosine-5 methyltransferases (DNMTs), DNMT1, DNMT3A, DNMT3B, and DNMT3C, have been identified in mammals and exhibit different functions in the mammalian methylation process [23]. In addition, non-canonical DNMTs consisting of DNMT2 and DNMT3L have also been identified [24,25]. DNMT3A and DNMT3B establish new DNA methylation patterns early in development and DNMT1 is responsible for the maintenance of established DNA methylation patterns [26,27]. In the context of its role in maintaining cell identity and induction of reprogramming, aberrant DNA methylation caused by mutation or overexpression of DNMT genes is observed in many cancers, including leukemias/lymphomas and lung, colon, pancreas, and breast cancers [23]. From these findings, DNMTs are considered as targets for cancer treatment, and nucleoside analogs, 5-azacytidine (azacytidine) and 5-aza-2-deoxycytidine (decitabine), which incorporate into DNA and inhibit DNMTs, are approved for the treatment of acute myeloid leukemia and myelodysplastic syndrome [28]. The cytotoxic effect of these drugs has also been determined in solid tumors, including breast cancers [29,30]. However, the existence of primary resistance and the emergence of secondary resistance, in which the drugs are initially effective, but tumors acquire resistance after long-term treatment, are obstacles in the treatment with these drugs [31]. While these drugs are effective as epigenetic modifiers when given at nanomolar doses, they induce DNA damage and cytotoxicity at high micromolar doses [32,33,34].

High levels of DNMT proteins correlate with effective responses to decitabine at nanomolar doses in patient-derived xenografts of TNBCs [34]. In addition, DNMT1 is considered to be involved in tumorigenesis of several types of cancer, leukemia/lymphoma, and multiple solid tumors, including TNBC [35]. Therefore, it is possible that the inhibition of DNMTs by decitabine is therapeutic by canceling the DNMT-mediated epigenetic inactivation of tumor suppressive factors, including apoptosis and cell cycle regulators. However, the mechanism underlying the decitabine-mediated anticancer effect and its efficacy in TNBC are still not fully understood. To clarify these issues, we analyzed TNBC cell lines for their sensitivity to decitabine and their expression of apoptosis-regulating molecules and examined the effect of combining decitabine with cisplatin on TNBC cells.

## 2. Materials and Methods

### 2.1. Cell Lines, Cell Culture, and Treatment

Human TNBC cell lines MBA-MB-468, MDA-MB-453, MDA-MB-231, HCC38, HCC1143, HCC1187, and HCC1937 were purchased from the American Type Culture Collection (Manassas, VA, USA). MDA-MB-157 and Hs578T cells were purchased from KAC Co., Ltd. (Kyoto, Japan). MBA-MB-468, MDA-MB-453, MDA-MB-231, and Hs578T cells were cultured in Dulbecco’s modified Eagle’s medium (DMEM) supplemented with 10% heat-inactivated fetal bovine serum (FBS; Nichirei, Tokyo, Japan) and 5 mM glutamine. HCC38, HCC1143, HCC1187, and HCC1937 cells were cultured in RPMI-1640 supplemented with 10% FBS and 1% sodium pyruvate. All cells were grown in humidified cell culture incubators under 5% CO_2_, 95% air at 37 °C.

All cell lines were seeded in plates before treatment. At 24 h after seeding, cells were treated with the indicated concentration of decitabine for 72 h and then used in experiments.

### 2.2. Antibodies

Antibodies were purchased as follows: BIM (C34C5), BCL-XL (54H6), PARP (#9542), Cleaved Caspase-3 (5A1E), DNMT1 (D63A6), DNMT3A (D23G1), and DNMT3B (E8A8A) from Cell Signaling Technology (Danvers, MA, USA); NOXA (114C307.1) from Thermo Fisher Scientific (Waltham, MA, USA); MCL1 (S-19), p21 (F-5) and ATF3 (C-19) from Santa Cruz Biotechnology (Santa Cruz, CA, USA); Anti-KLF4 (56CT5.1.6) from Bioss Inc. (Woburn, MA, USA); β-actin (AC-74) from Sigma-Aldrich (Tokyo, Japan); p27 (554069) and p16 (554079) from BD Pharmingen (San Jose, CA, USA); and ATF4 (739441) from R&D Systems (Minneapolis, MN, USA).

### 2.3. Compounds and Materials

Compounds were purchased as follows: decitabine from Wako Pure Chemical Industries, Ltd. (Osaka, Japan); cisplatin, doxorubicin and Q-VD-OPh from Calbiochem (San Diego, CA, USA); paclitaxel from Sigma-Aldrich; and hydroxychloroquine and necrostatin-1 from Cayman Chemical (Ann Arbor, MI, USA). Stock solutions of decitabine were prepared in dimethyl sulfoxide (DMSO; Sigma-Aldrich) at concentrations of 0.5 and 1 mM.

### 2.4. Cell Proliferation Assay

Cell proliferation assay was performed using the Cell Counting Kit-8 kit (CCK-8: Dojindo Molecular Technologies, Inc., Kumamoto, Japan) as described previously [36]. Briefly, TNBC cell lines were seeded in microtiter plates (96 wells, Greiner, Kremsmünster, Austria) at 5000 cells per well in 100 μL medium and treated with reagents for 72 h. After treatment, 10 μL of the CCK-8 reagent was added into each well and cells were incubated for 2 h at 37 °C. The optical density at 450 nm was measured using a SpectraMax 250 microplate reader (Molecular Probes, Eugene, OR, USA). The percentage of the treated group compared with that of the control was determined as cell viability.

### 2.5. Cell Death Lactate Dehydrogenase (LDH) Release Assay

Cell death LDH assay was performed using the Cytotoxicity LDH assay kit-WST (Dojindo Molecular Technologies, Inc, Kumamoto, Japan) as described previously [37]. Briefly, TNBC cell lines were seeded in 24-well plates at 5 × 10^4^ cells per well and incubated at 37 °C under 5% CO_2_ for 24 h before decitabine treatment. After treatment of 500 or 1000 nM decitabine for 72 h, the released LDH was measured using the kit with a SpectraMax 250 microplate reader (Molecular Probes).

### 2.6. Immunoblotting Analyses

Whole cell lysates were prepared with CHAPS lysis buffer [25 mM HEPES (pH 7.4), 250 mM NaCl, 1% CHAPS (3-[(3-cholamidopropyl) dimethylammonio]-1-propanesulfonate) containing protease inhibitor cocktail (Nacalai)]. For immunoblotting, equal amounts of proteins were loaded on SDS acrylamide gels, transferred to PVDF membranes, and analyzed by immunoblotting, as described previously [38].

### 2.7. Transcriptome Sequencing and Data Analysis

Total RNA was extracted from harvested HCC38 cells (>1.0 × 10^6^ cells) using the RNeasy Plus Mini Kit (Qiagen; Hilden, Germany). Transcriptome sequencing and bioinformatic analysis was performed by GENEWIZ, Inc. (Tokyo, Japan) using the Hiseq 2500 System (Illumina). Raw data were processed using the DESeq2 protocol for differential gene expression analysis. For further gene ontology, pathway, and enrichment analysis, Inguinal Pathway Analysis (IPA, Qiagen) was performed.

### 2.8. Bioinformatic Analysis of Methylation Levels

UALCAN (http://ualcan.path.uab.edu; accessed on 21 October 2021) is an online bioinformatics portal based on cancer data from The Cancer Genome Atlas database [39]. We used UALCAN to obtain promoter methylation levels of *PUMA*, *BIM*, *NOXA*, *BNIP3*, *p21*, *p27* and *p16* in normal and tumor tissues. The *p* value was calculated using Student’s *t*-test.

### 2.9. RNA Interference

Retrovirus-encoded short-hairpin RNAs (shRNAs), shNOXA, shBIM, and shPUMA, were cloned into the pSuper puro vector (Oligoengine; Seattle, WA, USA). The target sequences were as follows: 5′-GGAAACGGAAGATGGAATA-3′ (shNOXA#1), 5′-GCTACTCAACTCAGGAGAT-3′ (shNOXA#2), 5′-CTACCTCCCTACAGACAGA-3′ (shBIM), 5′-GGGTCCTGTACAATCTCAT-3′ (shPUMA). Lentiviral shRNA-expressing constructs were cloned into the plko.1 vector (Addgene; Cambridge, MA, USA). The target sequences were as follows: 5′-CCAGCCAGAAAGCACTACAAT-3′ (sh-KLF4), 5′-GAACTGCACTTCAGCAATAAT-3′(sh-BNIP3), and 5′-CCTAAGGTTAAGTCGCCCTCG-3′ (sh-control). The constructs were transfected into 293T packaging cells along with the packaging plasmids (Addgene) and the lentivirus-containing supernatants were used to transduce TNBC cells. Retroviral or lentiviral infection was performed as previously described [40]. Infected cells were selected using 1 μg/mL puromycin (Sigma-Aldrich) for 3 days.

siRNA transfections were performed using Opti-MEM and Lipofectamine RNAi Max (Invitrogen, Waltham, MA, USA) with a final siRNA concentration of 10 μM siRNA. Cells were transfected at a concentration of 10 nM for 24 h. The siRNAs used in this study were silencer negative control (4390843) and si-ATF4 (s1704) (Silencer Select siRNAs from Ambion, Austin, TX, USA).

### 2.10. Quantitative Real-Time PCR Analysis

Quantitative real-time PCR (qPCR) analysis was performed as described [41]. Predesigned primer/probe sets were used for analyses: β-actin, Hs03023880_g1; NOXA, Hs00560402_m1; BID, Hs00609632_m1; BAD, Hs00188930_m1; BIM, Hs00708019_s1; BIK, Hs00154189_m1; BNIP3, Hs0096929_m1 PUMA, Hs00248075_m1; p21, Hs00355782_m1; BAK, Hs00832876_gl; BAX, Hs00180269_mL; BCL-XL, Hs99999146_m1; MCL-1, Hs01050896_m1; ATF4, Hs00909569_g1; and KLF4, Hs00358836_m1. Data were calculated as mRNA levels relative to β-actin in accordance with the manufacturer’s protocol [41].

### 2.11. Cell Cycle Analysis

Cell cycle analysis was performed using the propidium iodide (PI)-staining method as described previously [36]. Briefly, cells were treated with decitabine at a concentration of 500 nM for 72 h and then harvested and fixed in cold 75% ethanol at 4 °C overnight. The fixed cells were collected, suspended in phosphate-buffered saline (PBS) buffer containing 10 μg/mL PI and 10 μg/mL RNase A, and incubated for 30 min at room temperature. DNA content was measured using a BD FACSCalibur (BD Biosciences, San Jose, CA, USA), and each histogram was constructed from data of at least 20,000 events. The data are expressed as percentages of total gated cells using Modfit LTTM Software (BD Biosciences).

### 2.12. Colony-Forming Assay

Each TNBC cell line (3 × 10^3^ cells/well) was seeded in 6-well plates and maintained in 5% CO_2_ at 37 °C. After 24 h, 500 nM decitabine and/or 20 µM cisplatin were added and cells were incubated for 7 days. Cells were then fixed with 4% paraformaldehyde at 25 °C for 10 min and stained using crystal violet solution (0.05% crystal violet, 1% formaldehyde, 1% methanol and 1× PBS) at 25 °C for 60 min. Stained cells were washed with water and air dried at room temperature. The number of colonies was quantified using a SpectraMax 250 microplate reader (Molecular Devices, San Jose, CA, USA) and the relative colony number is represented as the ratio relative to that of DMSO-treated cells.

### 2.13. Statistical Analysis

Data are shown as mean ± S.D. for three separate experiments. Student’s *t* test was used to analyze two-group comparisons. *p* values of less than 0.05 were considered statistically significant.

## 3. Results

### 3.1. Nanomolar Levels of Decitabine Induce Cell Death or Growth Inhibition in TNBC Cell Lines

First, we analyzed the cytotoxic effects of decitabine on nine TNBC cell lines. We used 500 and 1000 nM (1 μM) decitabine because these nanomolar doses of decitabine induced cell death in organoids from TNBC patients with high levels of DNMTs [34]. Moreover, previous studies showed that decitabine causes slight DNA damage at 1 μm and increased DNA damage at 10 μM in cancer cells [42] and 500 nM decitabine is widely used as an epigenetically targeted lower dose [43]. Cell viability and death responses to decitabine were measured 72 h after treatment. Cell death was determined by measuring the extracellular release of LDH [44], which is caused by both apoptosis and necrosis. As shown in Figure 1a,b, TNBC cell lines were classified into three types according to their sensitivity to decitabine: cell death (D)-type (MDA-MB-468 and HCC38), growth inhibition (G)-type (MDA-MB-453, MDA-MB-157, MDA-MB-231, and HCC1143), and resistant (R)-type (Hs578T, HCC1187, and HCC1937). In the D-type cells, a marked amount of cell death (more than 20% increase in LDH release) and a reduced number of viable cells were observed in response to decitabine. In contrast, G-type cells treated with decitabine showed a reduced number of viable cells, but only a small LDH release (less than 10%) was detected. In contrast, treated R-type cells showed neither a decrease in cell number nor an increase in LDH release. Moreover, decitabine-induced cell death in D-type cells, MDA-MB-468 and HCC38, was suppressed by a caspase inhibitor (Q-VD-OPh), as shown in Figure 1c,d, respectively. Caspases are key regulators and executers of apoptosis [32]. In contrast, decitabine-induced cell death was not affected by necrostatin-1, an inhibitor of necroptosis that enhances cell death caused by some anticancer drugs [45], or by chloroquine, an autophagy inhibitor that affects the induction of apoptosis by anticancer drugs [46]. These results indicate that decitabine induced apoptosis in TNBC cells. These findings were supported by decitabine only inducing cleavage of PARP and caspase-3 (Figure 1e), which is characteristic of apoptosis, in D-type cells MDA-MB-468 and HCC38 [47,48].

### 3.2. Decitabine Induces Gene Expression of Proapoptotic BCL2 Family Members and CDK Inhibitors

To understand the mechanism of decitabine-induced cell death and growth inhibition, we performed differential gene expression analysis and identified genes induced by decitabine in D-type HCC38 cells (Figure 2a,b). Among the apoptosis-regulated genes, the mRNA levels of proapoptotic BCL2 family proteins were induced by decitabine. BCL2 family proteins are key regulators of mitochondria-mediated apoptosis in response to extrinsic and intrinsic cell death signals [16]. Cell death signals initially induce pro-apoptotic BH3-only BCL2 family proteins, and induced BH3-only proteins activate pro-apoptotic members BAX and BAK, which function as apoptosis executer and inhibitor of anti-apoptotic members BCL2, BCL-XL, and MCL1 that constitutively inactivate BAX and BAK, resulting in induction of mitochondria-mediated apoptosis [16,49]. As shown in Figure 2a, decitabine induced the mRNA levels of the BH3-only proteins of the BCL2 family of apoptosis inducers [16], *PUMA*, *NOXA*, *BIM,* and *BMIP3*, but not of *BID* and *BAD*. Furthermore, among cell cycle regulators, the mRNA levels of *p27* and *p21*, cyclin-dependent kinase (CDK) inhibitors that inhibit cell growth [50], were induced by decitabine (Figure 2b).

Next, we analyzed the level of CpG methylation of these genes in breast cancers using the UALCAN database (http://ualcan.path.uab.edu; accessed on 21 October 2021). Hypermethylation of p16 is frequently observed in breast cancer [51]. As shown in Figure 2c, CpG methylation of CDK inhibitor genes, *p21*, *p27*, and *p16*, was relatively high in breast cancer tissues compared with that in normal tissues. In contrast, only CpG methylation of apoptosis inducer genes *BIM* and *BNIP3*, but not of *PUMA* and *NOXA*, was relatively high in breast cancer tissues (Figure 2c).

### 3.3. NOXA Is Induced by Decitabine in D-Type Cells

Having shown that decitabine induces apoptosis, we next analyzed the mRNA levels of BH3-only proteins in D-type cells by qPCR. In MDA-MB-468 cells, *BIM*, *BNIP3*, and *NOXA* mRNAs were induced by decitabine (Figure 3a). In contrast, *NOXA* was induced but lower levels of *BIM*, *BNIP3*, *BIK*, and *PUMA* mRNA were observed in HCC38 cells (Figure 3b). *NOXA* mRNA was analyzed in other cell lines, and the induction was only observed in G-type MDA-MB-157 cells (Figure 3c). We then analyzed NOXA and BIM protein levels in these cells. As shown in Figure 3d, NOXA and BIM proteins were induced in both G-type cells. In all cells, the levels of BCL-XL and MCL1 were not significantly affected by decitabine (Figure 3d). The role and mechanism of high NOXA levels in MDA-MB-231 cells was not further elucidated. However, NOXA is overexpressed in some cancer cells [52], indicating the existence of an inactivation system of the proapoptotic function of NOXA in these cells. In contrast to NOXA, BIM levels were constitutively observed and not markedly enhanced by decitabine (Figure 3d). In addition, decitabine-induced *NOXA* mRNA expression was observed in MDA-MB-157 cells (Figure 3c), but no NOXA protein expression (Figure 3d) and apoptosis induction (Figure 1b,e) was observed, suggesting that induction of apoptosis by NOXA is also regulated by post-transcriptional mechanism such as translation and stability of NOXA protein. These results indicate that NOXA plays a role in decitabine-induced apoptosis.

### 3.4. NOXA Regulates Decitabine-Induced Apoptosis in D-Type Cells

To understand the role of NOXA and other decitabine-inducible BH3-only proteins, PUMA, BNIP3, and BIM, we performed RNAi-mediated knockdown cells of these genes in MDA-MB-468 cells (Figure 4a). NOXA knockdown, but not PUMA, BIM, or BNIP3 knockdown, resulted in reduced cell death by decitabine (Figure 4b). Inhibition of apoptosis by NOXA knockdown was confirmed by the apoptosis markers cleaved PARP and cleaved caspase-3 (Figure 4c). The same antiapoptotic effect of NOXA knockdown was seen in another G-type cell line, HCC38 cells (Figure 4d–g). These results indicate that among the BH3-only proteins tested, only NOXA exerts an apoptosis-inducing function in response to decitabine in D-type cells.

### 3.5. Decitabine Induces the Expression of p21 and Inhibits Cell Cycle Progression in G-Type Cells

We next analyzed the expression of CDK inhibitors. The mRNA and protein levels of p21 were induced in G-type and D-type cells, but not in R-type cells (Figure 5a,b). The protein levels of other CDK inhibitors, p27 and p16, were not affected by decitabine. Moreover, we analyzed the cell cycle of G-type cells and found that decitabine mainly induced G2 cell cycle arrest, which is induced by p21, similar to G1 arrest [53] (Figure 5c).

### 3.6. Expression of DNMTs and Their Responsive to Decitabine in TNBC Cell Lines

The protein levels of DNMT1, DNMT3A, and DNMT3B correlate with the response to decitabine in TNBC patient-derived xenografts [34]. Moreover, decitabine induces degradation of DNMTs through TNF receptor-associated factor 6 (TRAF6), resulting in inhibition of cell growth [34]. However, as shown in Figure 6a (upper part), the levels of the three DNMT proteins in the TNBC cell lines varied and no characteristic expression pattern was observed among the D-, G-, and R-types. Analysis of two cell lines of each type showed that the protein levels of DNMT1 and DNMT3A were reduced by decitabine, but no obvious effect was observed on DNMT3B (Figure 6b). These results indicate that DNMT1 and DNMT3A are targets of decitabine in these cell lines. Moreover, DNMT1 knockout in cells induces aberrant expression in up to 10% of genes and tumor suppressor p53-dependent apoptosis [54]; however, in all TNBC cell lines, the p53 gene (*TP53*) is mutated (Figure 6c). These results indicate that the release of epigenetic restriction by decitabine induces cytotoxicity through p53-independent induction of regulators of apoptosis and cell growth in TNBC cells. *NOXA* and *p21* are p53 target genes [55,56], and the transcription factors activating transcription factor 3 and 4 (ATF3 and ATF4) and Krüppel-like factor 4 (KLF4) are also p53-independent transcriptional activators in response to DNA-damage [57,58]. However, decitabine did not induce the expression of ATF3, ATF4, or KLF4 (Figure 6b). In contrast, we found that among these transcription factors, the expression of NOXA in response to decitabine was suppressed by siRNA-mediated ATF4 knockdown in both D-type cells (Figure 6d). These results indicate that ATF4 is involved in the induction of NOXA expression in response to decitabine via a post-transcriptional mechanism. Notably, we previously reported that in some TNBC cells, NOXA is induced by KLF4 in a p53-independent manner in response to DNA damage [40]. However, KLF4 had no effect on the induction of NOXA by decitabine in these cells (Appendix A).

### 3.7. Decitabine Enhances the Cytotoxic Effect of Cisplatin in TNBC Cells

Cisplatin is an effective chemotherapeutic agent for TNBC, and the acquisition of resistance to cisplatin is an important prognostic issue [59]. Therefore, we examined the effect of cisplatin in combination with decitabine. As shown in Figure 7a, the cytotoxic effect of cisplatin was further enhanced by decitabine in D-type cells. Interestingly, cisplatin alone showed effective cytotoxicity on R-type cells, with no enhancing effect of decitabine (Figure 7a, see Discussion). In contrast, decitabine slightly enhanced cell growth inhibition by cisplatin after treatment for 72 h in G-type cells (Figure 7a). Suppressed effects of cell growth and survival were clearly observed in colony formation assays after seven days of treatment, and enhanced suppression was observed by the combined use of cisplatin with decitabine (Figure 7b). A similar decitabine-enhancing effect on G-type cells was also observed with doxorubicin, which elicits a DNA-damage response similar to that of cisplatin [60] (Figure 7c). However, the susceptibility of these cell lines to paclitaxel, another anticancer drug used in TNBC, was different from that of cisplatin, and no enhancing effect of decitabine was observed (Figure 7d; compared with Figure 7a).

### 3.8. Decitabine Augments the Cisplatin-Enhanced mRNA Levels of NOXA and Other BH3-Only Proteins

Finally, we analyzed whether the mRNA levels of BCL2 family proteins were enhanced by cisplatin in combination with decitabine in D-type cells. As shown in Figure 7a,b, only *NOXA* mRNA levels were synergistically enhanced by cisplatin and decitabine. Moreover, the cytotoxic effect of cisplatin with decitabine, but not cisplatin alone, was suppressed by NOXA knockdown (Figure 7c,d). Furthermore, in the G-type cells, MDA-MB-231 and HCC1143, decitabine also enhanced the induction of *NOXA* mRNA by cisplatin and induced mRNAs of many other BH3-only proteins (Figure 8e,f). These results indicate that the decitabine-enhanced cytotoxic effects of cisplatin are mediated in many, if not all cell lines, by the release of the epigenetic silencing of NOXA expression.

## 4. Discussion

We found that decitabine enhanced the proapoptotic effect of cisplatin on TNBC cell lines that are less sensitive to cisplatin through the induction of NOXA. These results indicate the potential for combination therapy with these drugs against TNBC. The DNA demethylating agents decitabine and 5-azacitidine are useful in the treatment of some cancers, especially acute myeloid leukemia and myelodysplastic syndrome [61]. The use of these drugs is currently the most common initial treatment in high-risk myelodysplastic syndrome patients who are ineligible for bone marrow transplantation, and approximately half of the treated patients show a hematological response, including some patients who show a complete response [62]. Although the use of DNA demethylating agents against solid tumors is not common at present, treatment with these agents in patients with cervical, ovarian, and colorectal cancers and pleural tumors has shown some benefit [29]. Furthermore, combination therapies of these agents with radiation, anticancer drugs, and immune checkpoint inhibitors are being investigated by experimental and clinical trials [63]. In most cancers, local hypermethylation occurs in 5–10% of the CpG islands in promoter regions, leading to silencing of important tumor suppressor genes [64]. For example, genes encoding the retinoblastoma tumor suppressor (*RB1*), cell cycle inhibitors p16^INK4a^ (*CDKN2A*) and p15^INK4a^ (*CDKN2B*), mismatch repair factor *MLH1*, inhibitory molecules for cancer invasion and metastasis such as E-cadherin (*CDH1*) and H-cadherin (*CDH13*), the apoptosis signal regulator *DAPK1*, and various transcription factors involved in tumor suppression are negatively regulated by DNA hypermethylation in cancer [65]. Among apoptosis-inducers, epigenetic silencing of *BIM* or *PUMA* has been reported in several cancers, including renal cell carcinoma and Burkitt lymphoma [16]. Therefore, the therapeutic effect of decitabine on cancer is thought to arise from the restoration of expression of tumor suppressors inhibited by promoter hypermethylation, which thereby suppresses the hallmarks of cancer, such as evading apoptosis, insensitivity to antigrowth signals and tissue invasion, and metastasis [13,14].

We found that decitabine induced apoptosis through the induction of NOXA in D-type cells. *NOXA* is a p53-target gene and NOXA regulates p53-dependent apoptosis [56,66], which is an important tumor suppressor mechanism of p53 [67]. We previously found that the induction of NOXA by p53 was suppressed in some colon cancer cells that show resistance to apoptosis by p53 overexpression and that lack NOXA inducibility by p53 was restored by 5-azacytidine [68]. These findings indicated the possibility that the expression of transcription factor(s) that induce *NOXA* is suppressed by methylation. However, we did not detect CpG methylation of the *NOXA* prompter in the previous study. Here, by querying the UALCAN database, we found that methylation levels around the *NOXA* gene were not high in normal or breast cancer tissues. We also found that decitabine augmented ATF4 transcriptional activation of the *NOXA* gene, indicating that genes involved in the regulation of *NOXA* mRNA induction are suppressed by DNA methylation. NOXA is an important regulator that mediates the cytotoxic effects of anticancer drugs, and cancer cells use several strategies to suppress the function of NOXA [69]. For example, the proteasomal degradation of NOXA protein is enhanced in chemotherapy-resistant cancer cells [70]. Moreover, histone deacetylase inhibitors reactivate epigenetically silenced *NOXA* gene expression and induce apoptosis in pancreas and lung cancers [71,72]. In TNBC cells, histone deacetylase inhibitors stimulate NOXA-mediated degradation of the antiapoptotic factor MCL1 and induce apoptosis [73]. NOXA is an MCL1-specific inhibitor [69]; therefore, it is possible that the NOXA pathway of MCL1 suppression is deficient in TNBCs. Related to this, high MCL1 levels are associated with poor prognosis in certain subtypes of breast cancer, including TNBC [74], and MCL1 knockdown induces apoptosis in a subset of TNBC cell lines [53]. In this context, the effect of the MCL1 inhibitor S63845 was investigated in TNBC cell lines and patient-derived xenografts with high levels of MCL1 [75]. These findings indicate that restoration of *NOXA* expression is useful for the treatment of TNBCs. We recently found that treatment of TNBC cells with a KLF4-inducing small compound, APTO-253, resulted in the induction of KLF4-mediated *NOXA* expression and NOXA-mediated apoptosis [40]. Here, we found that ATF4 induces *NOXA* expression in response to decitabine. We have not found any reports of ATF4 activation by decitabine, but we speculate it may be possible to develop treatments for TNBC by activating this pathway. Therefore, the restoration of *NOXA* expression by decitabine in this study may be useful strategy for the treatment of TNBC. However, it cannot be concluded from the present results whether NOXA is the only target for inhibition of apoptosis induction in TNBC cells. Indeed, decitabine or a combination of decitabine and cisplatin induced several proapoptotic factors in these cells, including high levels of *BIK* mRNA in MDA-MB-231 cells.

Similar to the NOXA gene, the expression of the p21 gene is also induced by decitabine in cancer cells, although its hypermethylation was not detected [76]. In this study, we found that the expression of the CDK inhibitor p21 was suppressed by methylation in G-type cells. The p21 gene is a well-known p53 target gene, and various other transcription factors such as SP1/SP3, AP2, SMADs, STATs, E2Fs, and CEBPα/β also activate p21 transcription [77,78]. Furthermore, the expression of p21 is regulated by BRCA1 in cooperation with p53, and BRCA1 is frequently mutated in breast cancer [79,80]. Therefore, it is possible that some of these transcription factors, or their complementary factors, are inactivated by over-methylation in cancer cells. Further analyses are needed to clarify these mechanisms.

We classified TNBC lines into three types on the basis of the effect of decitabine: cell death (D)-type, growth inhibition (G)-type, and resistant (R)-type. R-type cells responded well to cisplatin, and decitabine alone or in combination with cisplatin did not show any additional effects. These results indicate that although alteration of pharmacokinetic characteristics of decitabine cannot be completely ruled out, there are not many epigenetic changes in genes that are important for cancer development, including evading apoptosis in response to DNA damage, in R-type cells. The sensitivity of TNBCs against DNA-damaging agents, such as cisplatin, is relatively high because of their high incidence of DNA repair defects [6]. Indeed, R-type H1937 cells have a mutation in the *BRCA1* gene [81]. More TNBC cell lines need to be examined to confirm this notion. However, the NOXA-MCL axis is considered to be an important target of TNBC because NOXA knockdown reduced decitabine-induced apoptosis and MCL1 plays an important role in TNBC (see above). Moreover, in D- and G-type cells, we observed that decitabine inhibited cell growth and survival, respectively, effects that were enhanced in combination with cisplatin. Furthermore, NOXA induction by decitabine was observed in these cell types. Therefore, we speculate that NOXA reactivation by decitabine may be effective in the treatment of a broad range of TNBC subtypes.

TNBCs represent approximately 16% of total breast cancer cases. They occur frequently in young women and tend to exhibit aggressive, metastatic behavior [2,3]. As significant therapeutic molecular targets are not expressed in TNBCs, these patients tend to have poorer prognoses, further caused by the high metastatic progression [1]. At present, taxane and anthracycline are the mainstay treatments of early TNBCs, and several clinical trials have shown improved response rates with the addition of platinum agents [5]. Moreover, platinum agents including cisplatin are now used for high-risk patients, for example, patients with lymph node metastasis [5,6]. Therefore, the use of decitabine may be effective in increasing the effect of such platinum agents, and the combined use of decitabine with platinum agents may also be useful for treating TNBC that is resistant to other treatments. The present study did not investigate effects on tumors in vivo, nor did we investigate the effect on cancer stem cells or cancer immunity; therefore, determination of the effectiveness of decitabine is limited. However, by clarifying the effect of decitabine and the combination of decitabine and cisplatin at the cellular level and analyzing their underlying mechanism, we believe that these results are important for developing future TNBC treatments.

## 5. Conclusions

Our results show that decitabine enhances the proapoptotic effect of cisplatin on TNBC cell lines and that cisplatin alone is effective at inducing apoptosis in decitabine refractory subtypes, indicating the potential for combination therapy in TNBC.

## Figures and Tables

**Figure 1 cancers-14-00248-f001:**
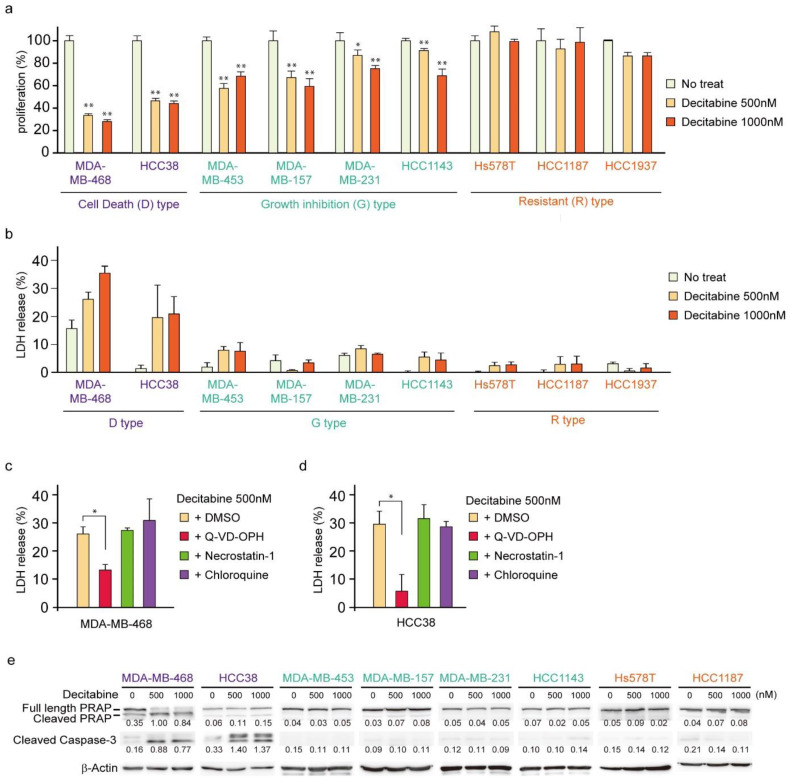
Effects of low-dose decitabine in nine TNBC cell lines. (**a**,**b**) Cells were exposed to the indicated amounts of decitabine (500 or 1000 nM). After treatment for 72 h, cell proliferation was measured by CCK-8 analysis (**a**) and cell death was measured by LDH release analysis (**b**). (**c**,**d**) MDA-MB-468 (**c**) and HCC38 (**d**) cells were treated with the indicated agents. After treatment for 72 h, cell death was determined using LDH release analysis. Error bars indicate SD (*n* = 3). The data are expressed as the mean ± SD of triplicate experiments. *, *p* < 0.05; **, *p* < 0.01. (**e**) Cells were exposed to the indicated amounts of decitabine (500 or 1000 nM). After treatment for 72 h, equal amounts of total cell lysate were subjected to immunoblot analysis with the indicated antibodies. Numbers below blots represent the relative band intensity of each protein normalized to the signal intensity of β-actin. The uncropped immunoblot images can be found in Appendix A.

**Figure 2 cancers-14-00248-f002:**
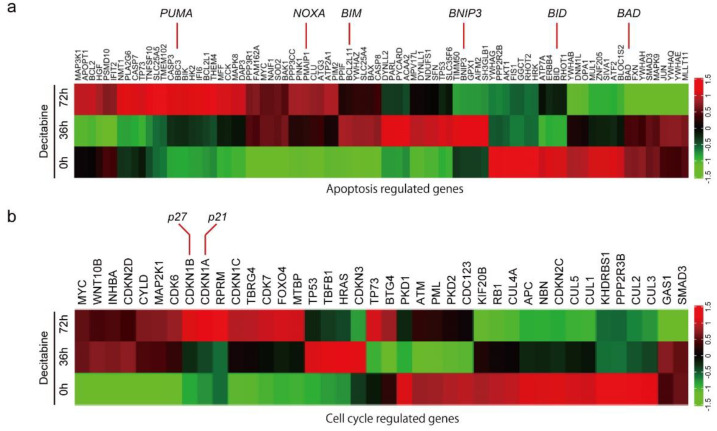
Differential gene expression and DNA methylation in mitochondrial apoptosis and cell cycle arrest genes in response to decitabine treatment. (**a**,**b**) RNA-seq expression analysis of HCC38 cells treated with vehicle or decitabine (500 nM) for 72 h. Heatmap of genes involved in (**a**) mitochondrial apoptosis and (**b**) cell cycle arrest. (**c**) The methylation level of each gene was examined by analyzing patient data extracted from The Cancer Genome Atlas database. *, *p* < 0.05.

**Figure 3 cancers-14-00248-f003:**
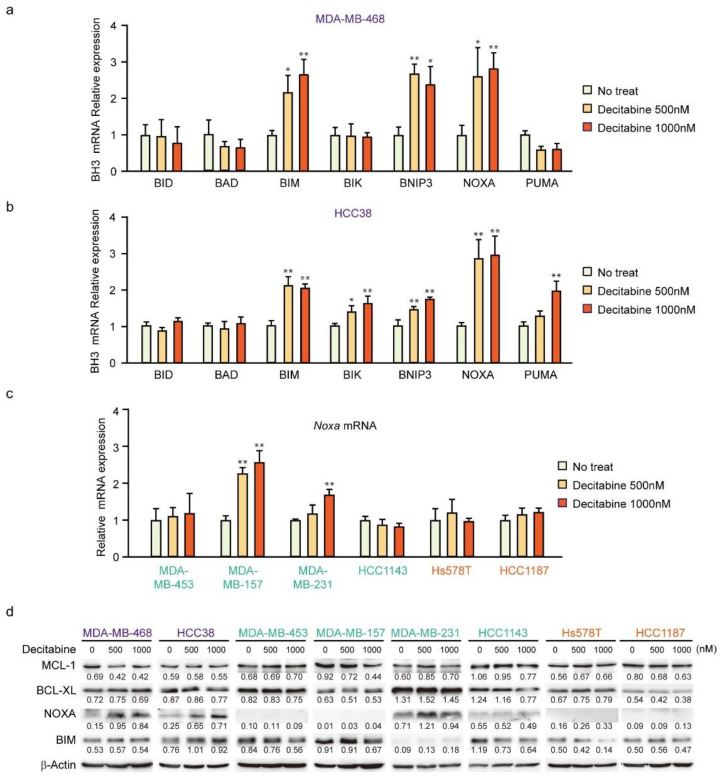
Expression of BCL2 family genes in TNBC cell lines. (**a**–**d**) Cells were exposed to the indicated amounts of decitabine (500 or 1000 nM). After treatment for 72 h, the mRNA levels in MBA-MB-468 (**a**) and HCC38 (**b**) cells were measured by quantitative real-time PCR. The levels of *NOXA* mRNA in indicated cells were determined by qPCR (**c**). Error bars indicate SD (*n* = 3). The data are expressed as the mean ± SD of triplicate experiments. *, *p* < 0.05; **, *p* < 0.01. (**d**) Equal amounts of total cell lysate were subjected to immunoblot analysis with the indicated antibodies. Numbers below blots represent the relative band intensity of each protein normalized to the signal intensity of β-actin. The uncropped immunoblot images can be found in Appendix A.

**Figure 4 cancers-14-00248-f004:**
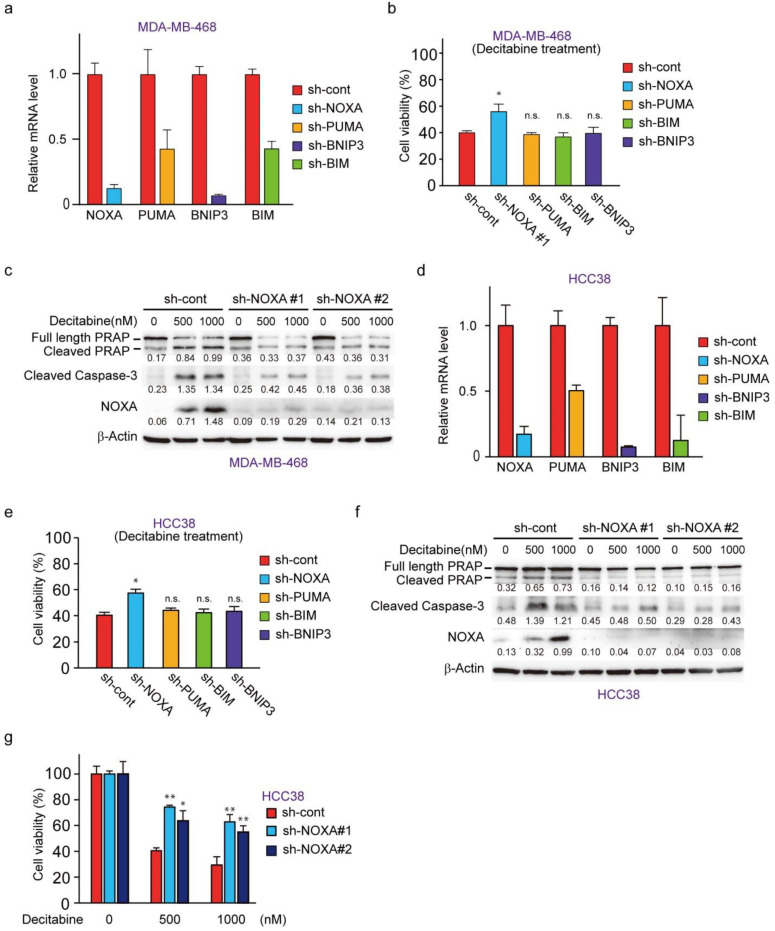
NOXA is a critical component for decitabine-induced apoptosis. (**a**) MBA-MB-468 cells stably expressing shRNA-targeting NOXA (sh-NOXA), PUMA (sh-PUMA), BNIP3 (sh-BNIP3), BIM (sh-BIM), or sh-control were assessed by qPCR. (**b**) MDA-MB-468 cells stably expressing each shRNA were exposed to decitabine (500 nM). After treatment for 72 h, cell viability was measured by CCK-8 analysis. (**c**) MBA-MB-468 cells stably expressing shRNA-targeting NOXA (sh-NOXA-#1, -#2) or sh-control were exposed to the indicated amounts of decitabine (500 or 1000 nM). After treatment for 72 h, equal amounts of total cell lysate were subjected to immunoblot analysis with the indicated antibodies. Numbers below blots represent the relative band intensity of each protein normalized to the signal intensity of β-actin. The uncropped immunoblot images can be found in Appendix A. (**d**) HCC38 cells stably expressing sh-NOXA, sh-PUMA, sh-BNIP3, sh-BIM, or sh-control were assessed by qPCR. (**e**) HCC38 cells stably expressing each shRNA were exposed to decitabine (500 nM). After treatment for 72 h, cell viability was measured by CCK-8 analysis. (**f**) HCC38 cells stably expressing shRNA-targeting NOXA (sh-NOXA) or sh-control were exposed to the indicated amounts of decitabine (500 or 1000 nM). After treatment for 48 h, equal amounts of total cell lysate were subjected to immunoblot analysis with the indicated antibodies. Numbers below blots represent the relative band intensity of each protein normalized to the signal intensity of β-actin. The uncropped immunoblot images can be found in Appendix A. (**g**) HCC38 cells stably expressing shRNA-targeting NOXA (sh-NOXA-#1, -#2) or sh-control were exposed to the indicated amounts of decitabine (500 or 1000 nM). After treatment for 72 h, cell viability was measured by CCK-8 analysis. Error bars indicate SD (*n* = 3). Data are expressed as the mean ± SD of triplicate experiments. *, *p* < 0.05; **, *p* < 0.01.

**Figure 5 cancers-14-00248-f005:**
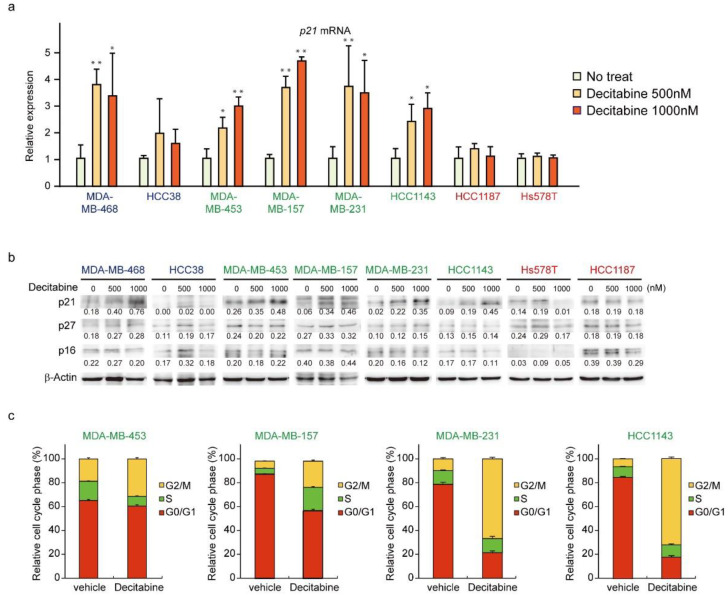
Decitabine induces cell cycle arrest through p21 induction. (**a**) Cells were exposed to the indicated amounts of decitabine (500 nM or 1000 nM). After treatment for 72 h, p21 mRNA was measured by quantitative real-time PCR. (**b**) Cells were exposed to the indicated amounts of decitabine (500 or 1000 nM). After treatment for 72 h, equal amounts of total cell lysate were subjected to immunoblot analysis with the indicated antibodies. Numbers below blots represent the relative band intensity of each protein normalized to the signal intensity of β-actin. The uncropped immunoblot images can be found in Appendix A. (**c**) Relative cell cycle phase was quantified by flow cytometry analysis of propidium iodide–stained cells. Error bars indicate SD (*n* = 3). The data are expressed as the mean ± SD of triplicate experiments. *, *p* < 0.05; **, *p* < 0.01.

**Figure 6 cancers-14-00248-f006:**
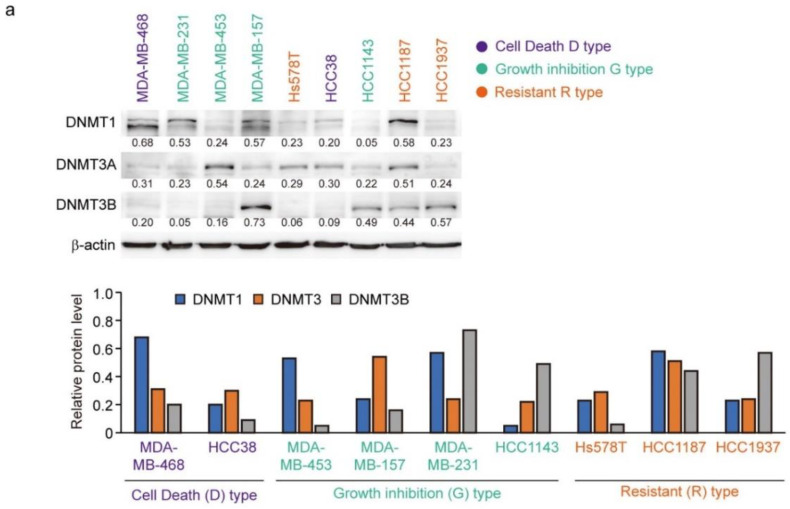
Levels of individual DNMT proteins do not correlate strongly with decitabine sensitivity. (**a**) Equal amounts of total cell lysate were subjected to immunoblot analysis with the indicated antibodies (**upper panel**). Protein expression levels were normalized to the loading control β-actin and fold change relative to the signal of loading control set to 1 (**lower panel**). (**b**) Each cell line was exposed to the indicated amounts of decitabine (500 or 1000 nM). After treatment for 72 h, equal amounts of total cell lysate were subjected to immunoblot analysis with the indicated antibodies. Numbers below blots represent the relative band intensity of each protein normalized to the signal intensity of β-actin (**upper panel**). Protein expression level were normalized to the loading control b-actin and fold change relative to the signal of loading control set to 1 (**lower panel**). The uncropped immunoblot images can be found in Appendix A. (**c**) Table showing the status of p53 in TNBC cell lines. (**d**) Cells were transfected with siRNAs (control-siRNA and siATF4). At 24 h after transfection, cells were exposed to 500 nM decitabine. After treatment for 72 h, mRNA levels in MBA-MB-468 (**left**) and Hcc38 (**right**) cells were measured by quantitative real-time PCR. Error bars indicate SD (*n* = 3). Data are expressed as the mean ± SD of triplicate experiments. *, *p* < 0.05; **, *p* < 0.01.

**Figure 7 cancers-14-00248-f007:**
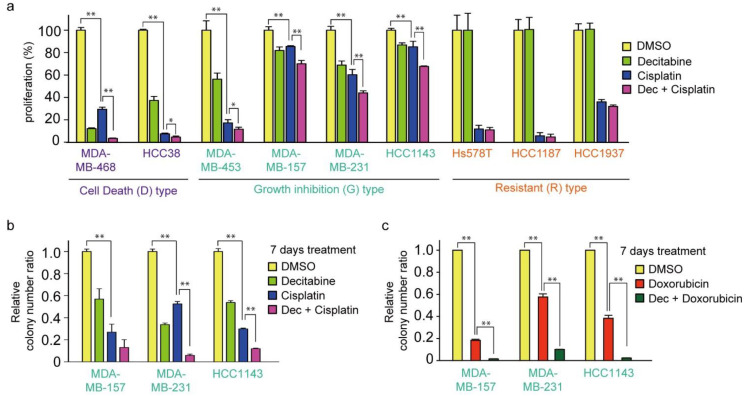
Decitabine enhances the cytotoxic effect of DNA-damaging agents in TNBC cells. (**a**) Each TNBC cell line was treated with vehicle DMSO (1 μg/mL), decitabine (500 nM), and/or cisplatin (20 μM). After treatment for 72 h, cell proliferation was measured by CCK-8 analysis. (**b**) Each cell line was treated with DMSO (1 μg/mL), decitabine (500 nM), and/or cisplatin (20 μM). After treatment for 7 days, relative colony numbers were quantified by microplate reader analysis of crystal violet–stained cells. (**c**) Each cell line was treated with DMSO (1 μg/mL), decitabine (500 nM), and/or doxorubicin (1 μg/mL). After treatment for 7 days, relative colony numbers were quantified by microplate reader analysis of crystal violet-stained cells. (**d**) Each TNBC cell line was treated with DMSO (1 μg/mL), decitabine (500 nM), and/or paclitaxel (20 nM). After treatment for 72 h, cell proliferation was measured by CCK-8 analysis. Error bars indicate SD (*n* = 3). Data are expressed as the mean ± SD of triplicate experiments. *, *p* < 0.05; **, *p* < 0.01.

**Figure 8 cancers-14-00248-f008:**
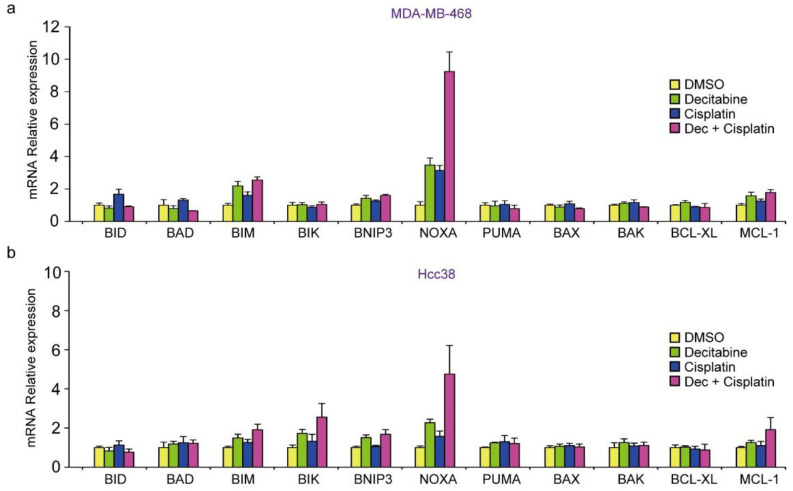
Decitabine enhances the cisplatin-induced mRNA levels of NOXA and other BH3-only proteins. (**a**,**b**) MDA-MB468 and HCC38 cells were treated with DMSO (1 μg/mL), decitabine (500 nM), and/or cisplatin (20 μM). After treatment for 72 h, the indicated mRNA levels in MBA-MB-468 (**a**) and HCC38 (**b**) cells were measured by quantitative real-time PCR. (**c**,**d**) MBA-MB-468 cells stably expressing shRNA-targeting NOXA (sh-NOXA) or sh-control were treated with DMSO (1 μg/mL), decitabine (500 nM), and/or cisplatin (20 μM). After treatment for 72 h, equal amounts of total cell lysate were subjected to immunoblot analysis with the indicated antibodies (**c**) and cell proliferation was measured by CCK-8 analysis (**d**). Numbers below blots represent the relative band intensity of each protein normalized to the signal intensity of β-actin. The uncropped immunoblot images can be found in Appendix A. (**e**) MDA-MB231 and HCC1143 cells were treated with DMSO (1 μg/mL), decitabine (500 nM), and/or cisplatin (20 μM). After treatment for 72 h, the indicated mRNA levels in MBA-MB-231 (e) and HCC1143 (**f**) cells were measured by quantitative real-time PCR. Error bars indicate SD (*n* = 3). Data are expressed as the mean ± SD of triplicate experiments. *, *p* < 0.05.

## Data Availability

Transcriptome sequences analyzed in this study are publicly accessible from the Gene Expression Omnibus (GEO) repository using accession number (currently registering).

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
