# Peer review of "Epigenetic Priming with Decitabine Augments the Therapeutic Effect of Cisplatin on Triple-Negative Breast Cancer Cells through Induction of Proapoptotic Factor NOXA"

_cancers, 2022, doi:10.3390/cancers14010248_

Round 1

Reviewer 1 Report

In the manuscript entitled "Epigenetic priming with decitabine augments the therapeutic effect of cisplatin on triple-negative breast cancer cells through induction of proapoptotic factor NOXA", the authors found that DNA-demethylating agent decitabine is therapeutically effective in triple-negative breast cancers (TNBCs), especially in combination with cisplatin. Since there are no effective molecular-targeted drugs for TNBCs, I found this paper interesting. However, some issues need to be considered and I have some question to authors:

  1. In Figure 3c, decitabine-induced enhancement of Noxa mRNA was observed in MDA-MB-157 cells to the same extent as MDA-MB-468 and HCC38. In Figure 1b, decitabine did not induce apoptosis in MDA-MB-157 cells, so the authors should explain this difference.
  2. In Figure 4d, the survival rate of cells in which each BH3-only protein was knocked down in HCC38 cells by decitabine treatment was measured in the same manner as in MDA-MB-468 cells shown in 4b. In Figure 4a, the authors showed that these BH3-only protein genes were knocked down in MDA-MB-468 cells by mRNA expression, so they should show the result that these genes are also knocked down in HCC38 cells.
  3. In Figure 6b, although expression of ATF3 protein was not detected, an increase in KLF4 protein by decitabine was clearly observed in MDA-MB-468 cells. In the authors' previous report, they showed that KLF4 induces NOXA expression in some TNBC cell lines (Nakajima, Genes 2021, 12, 539). Therefore, in order to elucidate the mechanism of NOXA induction by decitabine, I think that not only ATF4 but also KLF4 should be investigated.
  4. The authors showed in Figure 5 that p21 is induced in TNBC cell lines, which mostly carry the p53 mutant. They discuss the induction of p53-independent apoptosis in the Discussion, but not mentioned the induction of p53-independent p21. Therefore, I think they should discuss this.

Minor comments,

  1. Since there is large blank space on page 13, it is easier to see if the authors divide Figure 8 and fill in the gaps.
  2. I think it is necessary to add a Graphical Abstract to better understand the present results.

Author Response

We are grateful for the invaluable comments and suggestions made by the referees. In accordance with their suggestions, we have performed additional experiments to address the issues raised and have made amendments to the manuscript. Please find below our point-by-point responses to each of the comments.

In the manuscript entitled "Epigenetic priming with decitabine augments the therapeutic effect of cisplatin on triple-negative breast cancer cells through induction of proapoptotic factor NOXA", the authors found that DNA-demethylating agent decitabine is therapeutically effective in triple-negative breast cancers (TNBCs), especially in combination with cisplatin. Since there are no effective molecular-targeted drugs for TNBCs, I found this paper interesting. However, some issues need to be considered and I have some question to authors:

  1. In Figure 3c, decitabine-induced enhancement of Noxa mRNA was observed in MDA-MB-157 cells to the same extent as MDA-MB-468 and HCC38. In Figure 1b, decitabine did not induce apoptosis in MDA-MB-157 cells, so the authors should explain this difference.

--We think that this is an important point. As shown in Figure 3d, we found that NOXA protein expression is completely absent in MDA-MB-157 cells. These results suggest that NOXA expression is regulated by post-transcriptional mechanisms, such as regulation of translation and stability of NOXA, in addition to mRNA induction. Therefore, we have added the following comment into the text (line 315 to 319): “In addition, decitabine-induced NOXA mRNA expression was observed in MDA-MB-157 cells (Figure 3c), but no NOXA protein expression (Figure 3d) and apoptosis induction (Figure 1b and 1e) was observed, suggesting that induction of apoptosis by NOXA is also regulated by post-transcriptional mechanism such as translation and stability of NOXA protein.”

  1. In Figure 4d, the survival rate of cells in which each BH3-only protein was knocked down in HCC38 cells by decitabine treatment was measured in the same manner as in MDA-MB-468 cells shown in 4b. In Figure 4a, the authors showed that these BH3-only protein genes were knocked down in MDA-MB-468 cells by mRNA expression, so they should show the result that these genes are also knocked down in HCC38 cells.

--In accordance with this comment, we have added results showing that the genes for the BH3-only proteins were knocked down in HCC38 cells in Figure 4d.

  1. In Figure 6b, although expression of ATF3 protein was not detected, an increase in KLF4 protein by decitabine was clearly observed in MDA-MB-468 cells. In the authors' previous report, they showed that KLF4 induces NOXA expression in some TNBC cell lines (Nakajima, Genes 2021, 12, 539). Therefore, in order to elucidate the mechanism of NOXA induction by decitabine, I think that not only ATF4 but also KLF4 should be investigated.

--We agree with this comment and have performed additional experiments. These experiments revealed that KLF4 knockdown did not affect decitabine-induced NOXA expression in these cells. We have added this result in Supplemental S6 and added the following comments into the text (lines 396 to 398): “Notably, we previously reported that in some TNBC cells, NOXA is induced by KLF4 in a p53-independent manner in response to DNA damage [40]. However, KLF4 had no effect on the induction of NOXA by decitabine in these cells (Figure S6).”

  1. The authors showed in Figure 5 that p21 is induced in TNBC cell lines, which mostly carry the p53 mutant. They discuss the induction of p53-independent apoptosis in the Discussion, but not mentioned the induction of p53-independent p21. Therefore, I think they should discuss this.

--In accordance with this comment, we have added the following text into the Discussion (line 531 to 540): “Similar to the NOXA gene, the expression of the p21 gene is also induced by decitabine in cancer cells, although its hypermethylation was not detected [76]. In this study, we found that the expression of the CDK inhibitor p21 was suppressed by methylation in G-type cells. The p21 gene is a well-known p53 target gene, and various other transcription factors such as SP1/SP3, AP2, SMADs, STATs, E2Fs and CEBPα/β also activate p21 transcription [77,78]. Furthermore, the expression of p21 is regulated by BRCA1 in cooperation with p53, and BRCA1 is frequently mutated in breast cancer [79,80]. Therefore, it is possible that some of these transcription factors, or their complementary factors, are inactivated by over-methylation in cancer cells. Further analyses are needed to clarify these mechanisms.”

Minor comments,

  1. Since there is large blank space on page 13, it is easier to see if the authors divide Figure 8 and fill in the gaps.

--In response to this comment, we have changed Figures 1, 2, 6 and 8 to reduce blank spaces.

  1. I think it is necessary to add a Graphical Abstract to better understand the present results.

--In accordance with this comment, we have added a Graphical Abstract.

Reviewer 2 Report

The study entitled “Epigenetic priming with decitabine augments the therapeutic effect of cisplatin on triple-negative breast cancer cells through induction of proapoptotic factor NOXA” by Dr. Nakajima analyzed the effects of hypomethylating agent decitabine on nine TNBC cell lines and analyzed genes associated with its cytotoxic effects. The main results indicate that decitabine enhanced the proapoptotic effect of cisplatin on triple-negative breast cancer cell lines that are less sensitive to cisplatin through the induction of proapoptotic factor NOXA. Moreover, cisplatin alone is effective at inducing apoptosis in decitabine refractory triple-negative breast cancer cell subtypes, indicating the potential for combination therapy. The ms is in general well written and sections well organized, while it might reach interest across the readers as reporting very interesting findings with potential clinical application for combination cancer therapy. 

While recommending a minor revision, I have a several observations for improving the ms

General comments
•    The mmethos section is completely lacking in supporting references. Please include supporting references. In addition, as describing the  compounds employed, one of the methods’ subhead can be modified as “Antibodies, compounds and Materials”
•    Please include details on the drug treatments, including schedule
•    The rational behind the selection of 500 and 1000 nM concentration for decitabine should be more deeply detailed. Have the IC50 been determined for decitabine in the nine TNBC cell lines?
•    Figure 2, panel c, please include the database name in the figure caption
•    The quality of figures should be improved, especially, figure 6

Minor observations
Lines 81-83 Additional DNMTs have been described. Four members of DNMT family, including DNMT1, DNMT3A, DNMT3B, and DNMT3C, present catalytic activity (PMID: 28127428). Additional, non-canonical, DNMTs comprise DNMT2 and DNMT3L (PMID: 27232191 and PMID: 20838592). Please include these notions as a background

Line 89-91. Although helpful, the use of HMAs such as Decitabine and Azacitidine inevitably leads to primary and secondary resistance to this treatment (PMID:33958699). For completeness, this information should be included.

LIne 386 the dMSO condition should be described in figure 7 caption

Author Response

We are grateful for the invaluable comments and suggestions made by the referees. In accordance with their suggestions, we have performed additional experiments to address the issues raised and have made amendments to the manuscript. Please find below our point-by-point responses to each of the comments.

The study entitled “Epigenetic priming with decitabine augments the therapeutic effect of cisplatin on triple-negative breast cancer cells through induction of proapoptotic factor NOXA” by Dr. Nakajima analyzed the effects of hypomethylating agent decitabine on nine TNBC cell lines and analyzed genes associated with its cytotoxic effects. The main results indicate that decitabine enhanced the proapoptotic effect of cisplatin on triple-negative breast cancer cell lines that are less sensitive to cisplatin through the induction of proapoptotic factor NOXA. Moreover, cisplatin alone is effective at inducing apoptosis in decitabine refractory triple-negative breast cancer cell subtypes, indicating the potential for combination therapy. The ms is in general well written and sections well organized, while it might reach interest across the readers as reporting very interesting findings with potential clinical application for combination cancer therapy. 

While recommending a minor revision, I have a several observations for improving the ms

General comments
•    The mmethos section is completely lacking in supporting references. Please include supporting references. In addition, as describing the  compounds employed, one of the methods’ subhead can be modified as “Antibodies, compounds and Materials”.

--In accordance with this comment, we have separated “2.2. Antibodies and Materials” into “2.2. Antibodies” and “2.3. Compounds and materials.” Moreover, we have added supporting references into sections 2.5, 2.8 and 2.9.

  •    Please include details on the drug treatments, including schedule

--In accordance with this comment, we have added the following comment into the Materials and Methods (line 124 to 126): “All cell lines were seeded in plates before decitabine treatment. After seeding 24h, cells were treated with the indicated concentration of decitabine for 72h and performed each experiment,” and Figure 3 legend: “Cells were exposed to the indicated amounts of decitabine (500 nM or 1000 nM). After treatment for 72 h, the mRNA levels in MBA-MB-468 (a) and HCC38 (b) cells were measured by quantitative real-time PCR (qPCR).”

  •    The rational behind the selection of 500 and 1000 nM concentration for decitabine should be more deeply detailed. Have the IC50 been determined for decitabine in the nine TNBC cell lines?

--In accordance with this comment, we have added the following information in the manuscript (line 231 to 236): “First, we analyzed the cytotoxic effects of decitabine on nine TNBC cell lines. We used 500 nM and 1000 nM (1 μM) decitabine because these nanomolar doses of decitabine effectively induce cell death in organoids from TNBC patients with high levels of DNMTs [34]. Moreover, it has been shown that decitabine causes slight DNA damage at 1 μM and obvious DNA damage at 10 μM in cancer cells [42], and that 500nM decitabine is widely used as epigenetically targeted lower dose [43].”

  • Figure 2, panel c, please include the database name in the figure caption results.

  --In accordance with this comment, we have changed the legend for Figure 2c as follows (line 300 to 301): “The methylation level of each gene was examined by analyzing patient data extracted from The Cancer Genome Atlas database.”

  •    The quality of figures should be improved, especially, figure 6

--As the reviewer pointed out, the diagram was confusing and difficult to understand. We showed the band intensity in this figure, but we understand that it was confusing to interpret. Therefore, we changed the figure and have shown the band density in graph format in Figure 6. We hope that our revised figure is easier to interpret.

 Minor observations
Lines 81-83 Additional DNMTs have been described. Four members of DNMT family, including DNMT1, DNMT3A, DNMT3B, and DNMT3C, present catalytic activity (PMID: 28127428). Additional, non-canonical, DNMTs comprise DNMT2 and DNMT3L (PMID: 27232191 and PMID: 20838592). Please include these notions as a background

--In accordance with this comment, we have added the following information into the text regarding these DNMTs (line 79 to 85): “In eukaryotic genomes, methylation of cytosines within CpG dinucleotides is involved in transcriptional silencing and maintaining and regulating chromatin structure, genome stability and cell fate [21,22]. Four cytosine-5 methyltransferases (DNMTs), DNMT1, DNMT3A, DNMT3B and DNMT3C have different functions in the mammalian methylation process [23]. In addition, non-canonical DNMTs consisting of DNMT2 and DNMT3L have also been identified [24,25].”

Line 89-91. Although helpful, the use of HMAs such as Decitabine and Azacitidine inevitably leads to primary and secondary resistance to this treatment (PMID:33958699). For completeness, this information should be included.

--In accordance with this comment, we have added the following information into the text (line 96 to 99): “However, the existence of primary resistance and the emergence of secondary resistance, in which the drugs are initially effective but tumors acquire resistance after long-term treatment, are obstacles in the treatment with these drugs [31].”

LIne 386 the dMSO condition should be described in figure 7 caption

--In accordance with this comment, we have added the following information into the legend of Figure 7: “Decitabine enhances the cytotoxic effect of DNA-damaging agents in TNBC cells. (a) Each TNBC cell line was treated with drug vehicle DMSO (1 μg/ml), decitabine (500 nM) and/or cisplatin (20 μM). After treatment for 72 h, cell proliferation was measured by CCK-8 analysis. (b) Each cell line was treated with DMSO (1 μg/ml), decitabine (500 nM) and/or cisplatin (20 μM). After treatment for 7 days, relative colony numbers were quantified by microplate reader analysis of crystal violet-stained cells. (c) Each cell line was treated with DMSO (1 μg/ml), decitabine (500 nM) and/or doxorubicin (1 μg/ml). After treatment for 7 days, relative colony numbers were quantified by microplate leader analysis of crystal violet-stained cells. (d) Each TNBC cell line was treated with DMSO (1 μg/ml), decitabine (500 nM) and/or paclitaxel (20 nM). After treatment for 72 h, cell proliferation was measured by CCK-8 analysis.”

We also included the following information for the legend for Figure 8: “Decitabine enhances the cisplatin-induced mRNA levels of NOXA and other BH3-only proteins. (a and b) MDA-MB468 and HCC38 cells were treated with DMSO (1 μg/ml), decitabine (500 nM) and/or cisplatin (20 μM). After treatment for 72 h, the indicated mRNA levels in MBA-MB-468 (a) and HCC38 (b) cells were measured by quantitative real-time PCR (qPCR). (c and d) MBA-MB-468 cells stably expressing shRNA-targeting NOXA (sh-NOXA) or sh-control, were treated with DMSO (1 μg/ml), decitabine (500 nM) and/or cisplatin (20 μM). After treatment for 72 h, equal amounts of total cell lysate were subjected to immunoblot analysis with the indicated antibodies (c) and cell proliferation was measured by CCK-8 analysis (d). Numbers below blots represent the relative band intensity of each protein normalized to the signal intensity of β-actin. The uncropped immunoblot images can be found in Figure S7. (e) MDA-MB231 and HCC1143 cells were treated with DMSO (1 μg/ml), decitabine (500 nM) and/or cisplatin (20 μM). After treatment for 72 h, the indicated mRNA levels in MBA-MB-231 (e) and HCC1143 (f) cells were measured by quantitative real-time PCR (qPCR).”